# Luminescent Zn Halide Complexes with 2-(2-Aminophenyl)benzothiazole Derivatives

**Taisiya S. Sukhikh** [1,*] [ID]**, Dmitry S. Kolybalov** [1,2] [ID]**, Ekaterina K. Pylova** [1,2,3] **and Sergey N. Konchenko** [1,2]

1   Nikolaev Institute of Inorganic Chemistry, Siberian Branch, Russian Academy of Sciences, Novosibirsk 630090, Russia
2   Department of Natural Sciences, National Research University—Novosibirsk State University, Novosibirsk 630090, Russia
3   Institute Charles Gerhardt Montpellier, National School of Chemistry Montpellier, University of Montpellier, CNRS, ENSCM, 34000 Montpellier, France
*   Correspondence: sukhikh@niic.nsc.ru

**Abstract:** We report a comparative study of coordination behaviour of 2-(2-aminophenyl)benzothiazole (NH$_2$-*pbt*) and its phosphorus-containing derivative, α-aminophosphine oxide (PCNH-*pbt*), towards zinc halides. The corresponding coordination compounds [Zn(L)$_2$Hal$_2$] (L = PCNH-*pbt*, Hal = Cl, **1** and Hal = Br, **2**) and [Zn(L′)Hal$_2$] (L′ = NH$_2$-*pbt*, Hal = Cl, **3** and Hal = Br, **4**) were obtained as single phases. As evidenced by single-crystal X-ray diffraction analysis, L′ ligand coordinates to Zn in a chelate manner via two N atoms. Despite a similar coordination mode in complexes **3** and **4**, the spatial geometry of the ligand differs notably, which implies a relatively high flexibility of NH$_2$-*pbt*. The L ligand exhibits another coordination mode, binding with Zn only via the oxygen of the P=O group. The differences in the structures of NH$_2$-*pbt*, **3** and **4**, and their counterparts, PCNH-*pbt*, **1** and **2,** induce differences in their solid-state photoluminescence properties. The former group of the compounds exhibits conventional single-band emission, while the latter group reveals two bands. The minor band at 450 nm is ascribed to a radiative transition for the regular amine species, while the major band at 520–550 nm can be associated either with the proton-transferred imine species (ESIPT mechanism) or with a charge transfer state (TICT) with a different geometry.

**Keywords:** coordination compounds; heterocyclic compounds; single crystal X-ray diffraction; luminescence

## 1. Introduction

The design of novel photoactive heterocyclic compounds is one of the hottest trends in view of the widest applications of these compounds in various fields. Heterocyclic derivatives can be used to create materials for organic light-emitting diodes (OLEDs) [1,2], sensors [3–6], solar energy converters [7,8], catalysts [9,10], etc. Recent studies of organic compounds based on 2-(2′-aminophenyl)benzothiazole (NH$_2$-*pbt*) have revealed their high potential in various photophysical applications, particularly owing to the presence of excited-state intramolecular proton transfer (ESIPT) [11–14]. At the same time, the coordination chemistry of NH$_2$-*pbt* and its derivatives is poorly studied; to the best of our knowledge, it is limited to the synthesis of a few complexes. Among them, only two Re complexes with NH$_2$-*pbt* [15,16] have been structurally characterized by means of single crystal X-ray diffraction. Examples of structurally characterized compounds with a deprotonated ligand include one Pd complex with the derivative bearing a monothiooxamide group [17] and two Re complexes with NH-pbt⁻ [18]. A series of papers [1,2,19] describe the intriguing exciplex-featured electroluminescence of devices with *pbt*-based sulfonamide complexes as a light-emitting layer. NH$_2$-*pbt* derivatives bearing acetamide groups have been studied as fluorescent probes for Zn$^{2+}$ ions in liquid media [20,21]; complexation with zinc was proposed as a sensing mechanism, but the corresponding complexes have not been isolated in their individual form.

With the aim of deepening the coordination chemistry of NH$_2$-*pbt* derivatives, we describe the synthesis and crystal structure of novel Zn halide coordination compounds with NH$_2$-*pbt* as well as with the α-aminophosphine oxide comprising the *pbt* moiety (PCNH-*pbt*) and study their photophysical properties.

## 2. Results and Discussion

### 2.1. Syntheses and Crystal Structures

1,3-Aminophosphine PCNH-*pbt* was synthesized from NH$_2$-*pbt* using the phospha-Mannich reaction as reported recently [22]. Interaction between ZnHal$_2$ (Hal = Cl and Br) and PCNH-*pbt* in ethyl acetate results in precipitation of compounds [Zn(L)$_2$Hal$_2$] (**1** and **2**, correspondingly; Scheme 1). The use of a stoichiometric ratio of the reagents (1:2) results in a mixture of solid phases. According to powder XRD analysis (Figures S1 and S2), the mixture consists of the title products and PCNH-*pbt* in a molar ratio of 55/45% for compound **1** and 40/60% for **2**. The phases exhibit approximately equal solubility in common organic solvents, which does not allow us to separate compounds **1** and **2** from free PCNH-*pbt* by a washing procedure. Pure phases **1** and **2** were obtained in the reactions with an excess of ZnHal$_2$, i.e., at a molar ratio of the reagents of ca. 1:1 (Figures S3 and S4). For comparison, we performed reactions between archetypal NH$_2$-*pbt* and ZnHal$_2$ to give [Zn(L')Hal$_2$] (Hal = Cl, **3** and Hal = Br, **4**). The compounds under study were characterized by single crystal and powder XRD analysis and IR spectroscopy. Crystals of **1** suitable for single crystal XRD can be prepared by a concentration of the mother liquor, while compound **2** is formed as a fine powder; its isostructurality with **1** has been determined by powder XRD analysis (Figure S4). Once solids **1** and **2** precipitate from the mother liquor, they become poorly soluble in organic solvents, which does not allow us to study them by means of NMR spectroscopy. In highly polar solvents, the compounds likely dissociate, and thus we did not study their photophysical properties in the solution.

**Scheme 1.** Synthesis of compounds **1**–**4**.

Single-crystal XRD analysis of compounds **3** and **4** revealed that the L' ligand coordinates to the metal in a chelate manner via N atoms of the amino group and the heterocycle moiety (Figure 1), as in previously reported Re complexes [15,16]. Despite the similar coordination mode, the ligand reveals different spatial geometries: in **3**, all non-hydrogen atoms of NH$_2$-*pbt* lie approximately in the same plane, while complex **4** features a large twist angle (of 26.7°) between heterocyclic and aniline moieties (Table S1). The reason for

the obvious unfavourable twisted conformation of NH$_2$-*pbt* in **4** is unclear: the bromide ligands do not produce apparent hindrance to the flattened conformation. Probably, the reason lies in different crystal packing effects in **3** and **4** (Figure S5). For **3**, hydrogens of the amino group are arranged at both sides of the *pbt* plane and are engaged in intermolecular hydrogen bonds with Cl atoms, forming a chain-like motif with the basic graph set [23] $C_2^2(6)$. In the case of **4**, both hydrogens are arranged on one side of the phenyl plane, forming a hydrogen-bonded motif with only one Br atom and featuring the most distinctive graph set $C_4^2(8)$. Through the hydrogen bonds, the molecules of **4** are connected in a chain with the topology $C_3^2(8)$. Thus, the structure of complex **4** demonstrates the relatively high flexibility of phenylbenzothiazole towards twist along the C–C bond despite the presence of conjugation between the heterocycle and aniline moieties. Since both N atoms participate in the coordination, the complexes do not show the intramolecular hydrogen bond typical for NH$_2$-*pbt* derivatives.

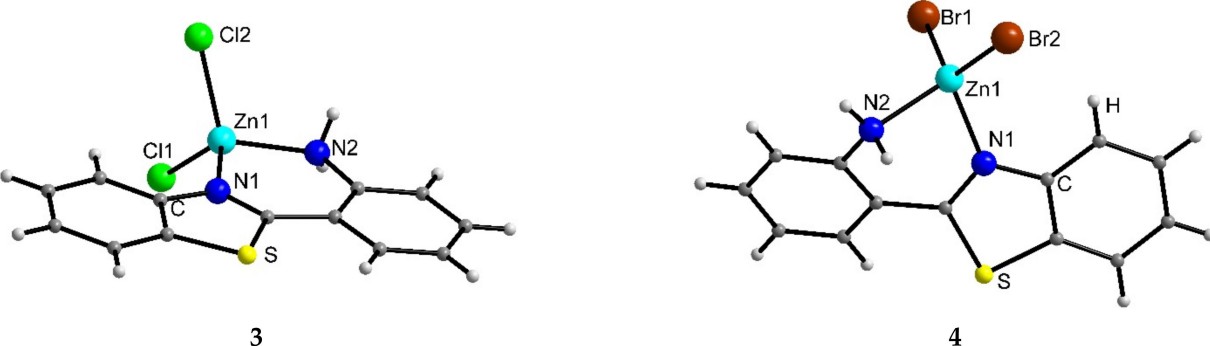

**Figure 1.** The structure of compounds **3** and **4**. Atomic color scheme: turquoise—Zn, green—Cl, brown—Br, blue—N, yellow—S, dark gray—C, light gray—H. Selected bond lengths (Å) for **3**: Zn1–Cl1 2.2195(5), Zn1–Cl2 2.2310(5), Zn1–N1 2.0144(15), Zn1–N2 2.0139(17). Selected bond lengths (Å) for **4**: Br1–Zn1 2.3764(3), Br2–Zn1 2.3419(3), Zn1–N1 2.0707(16), Zn1–N2 2.0622(17).

Powder XRD analysis revealed the isostructurality of compounds **1** and **2**. The structure of **2** was determined by single-crystal XRD analysis, while Rietveld refinement for **1** using the model of **2** with Br atoms replaced by Cl atoms, gave reasonable agreement with the experiment (Figure S2). Contrary to compounds **3** and **4**, the ligands in **1** and **2** do not coordinate via the N atoms but coordinate in a monodentate fashion via the O atom of the P=O group (Figure 2a). The different coordination behaviour of L and L' probably consists of steric hindrance of diphenylphosphine and furanyl groups in the latter, which does not allow coordination to Zn in a chelate N,N or N,N,O fashion.

Both ligands in **2** feature typical twist angles of 3.6 and 15.0° between the heterocyclic and aniline moieties (Table S1). The compound reveals an intramolecular N–H⋯N hydrogen bond in the NH-*pbt* moiety. The geometry characteristics of the bond are also comparable with those for relative derivatives comprising the NH-*pbt* moiety [22,24,25] (Table S1).

PCNH-*pbt* molecules possess chirality at the backbone C atom; since the starting compound PCNH-*pbt* comprises both enantiomers, the chirality of two ligands in complex **2** in the solution is random. In the solid state, the ligands predominately possess similar chirality (*R,R* or *S,S*) in a molecule of **2**. However, the crystal packing can be violated by the presence of another molecule, *R,S-* or *S,R*-diastereomer. The geometric shape of both enantiomers in the complex is similar since the phenyl and furanyl groups attached to the backbone C atom overlay well with each other. The crystal packing violation is manifested by the presence of a disorder of {PPh} and {C(furanyl)} moieties of one ligand and phenyls of another ligand over two positions with an occupancy of 74/26% (Figure 2b). The structure of free PCNH-*pbt* reveals a similar disorder but to a lesser extent [22].

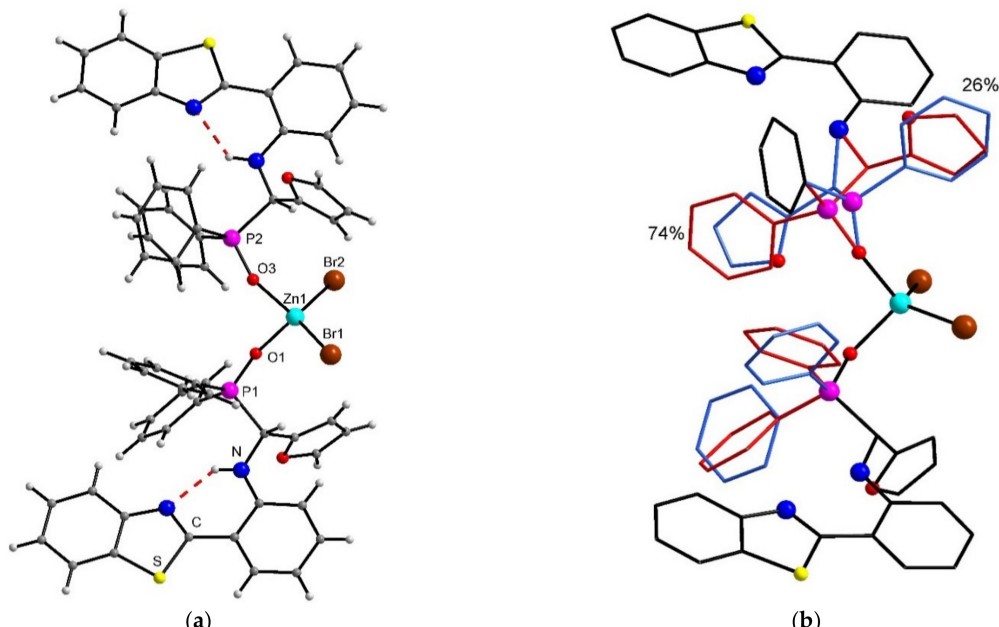

(**a**)　　　　　　　　　　　　　　　　　　　　　　(**b**)

**Figure 2.** (**a**) The structure of compound **2**; disorder is omitted, and the N–H···N hydrogen bond is marked by a dashed red line. Atomic color scheme: turquoise—Zn, brown—Br, magenta—P, red—O, blue—N, yellow—S, dark gray—C, and light gray—H. (**b**) The structure of compound **2** shows disorder; disordered moieties are red or blue, and hydrogen atoms are omitted. Selected bond lengths (Å) for **2**: Br1–Zn1 2.3464(6), Br2–Zn1 2.3567(6), Zn1–O1 1.969(2), Zn1–O3 1.982(3), P1–O1 1.508(2), and P2–O3 1.539(3).

## 2.2. Photoluminescence Properties

As in the case of free PCNH-*pbt* [22], solid compounds **1** and **2** reveal dual-band photoluminescence with maxima at 450 and 600 nm (Figure 3a). The first band is similar, while the second one is hypsochromically shifted by 10 nm as compared to PCNH-*pbt*. The presence of two bands can be a consequence of the existence of several excited states capable of radiative transitions. Using quantum chemical calculations for related aminophosphines based on *pbt*, we previously suggested possible pathways: (1) through states with different molecular geometry, viz., locally excited state (LE) and a charge transfer state (TICT); (2) via excited-state intramolecular proton transfer (ESIPT) [24]. These pathways can also occur in the compounds under study. Contrary to the aminophosphines, the long wavelength band always dominates in the cases of **1** and **2**, while the ratio of the bands negligibly changes in the excitation wavelength range of 300–380 nm. The excitation spectra resemble each other except for the somewhat different shape of the band at an emission wavelength of 440 nm as compared to those of 550–600 nm (Figure S6a). After a time delay of 100 µs, we detected no emission signal above the noise in the entire studied wavelength range, which indicates relatively short emission lifetimes of the compounds.

Interestingly, solutions of PCNH-*pbt* in tetrahydrofuran ($2 \cdot 10^{-5}$ M and $10^{-3}$ M) and toluene ($2 \cdot 10^{-5}$ M) reveal only the short wavelength band (Figure S6b). Addition of an excess of triethylamine has no effect on the emission. Thus, the long wavelength is attributed to some solid-state effects. Since no strong intermolecular interactions of the same nature are observed in crystals of PCNH-*pbt*, **1** and **2**, we can assume that the long wavelength band appears in the solid PCNH-*pbt* owing to geometry differences of the molecule in the solid state and solution [26]. The restriction of intramolecular motion (RIM) in the solid could also be responsible for the band appearance [27]. The results do not directly indicate the presence of ESIPT or TICT, but they do not contradict each other.

For comparison, we studied the solid-state photoluminescence of compounds **3** and **4** (Figure 3b). The emission spectra exhibit a major band at 395 and 407 nm for **3** and **4**, respectively, which is significantly hypsochromically shifted as compared to free NH$_2$-*pbt*

(455 nm). Both **3** and **4** reveal a minor band at ca. 520 and 550 nm; it likely arises from the presence of some defects or impurities that cannot be detected by other physical-chemical methods (powder XRD, elemental analysis, and NMR spectroscopy). We exclude that the minor band is associated with intrinsic features of the complexes since its relative intensity varies from sample to sample (Figure S7a). After a time delay of 100 μs, the emission spectra reveal only the long wavelength band (Figure S7b), which indicates its phosphorescent nature.

The major band for PCNH-*pbt* and its coordination compounds is located at much longer wavelengths than that for NH$_2$-*pbt*, **3** and **4**. This further suggests that the minor band at 450 nm can be ascribed to a radiative transition for the regular amine species (Scheme 2), while the major band at 520–550 nm is associated either with the proton-transferred imine species (ESIPT mechanism) or charge transfer state (TICT) with a different geometry. As evidenced by quantum chemical calculations [28], the ESIPT process does not occur in NH$_2$-*pbt*.

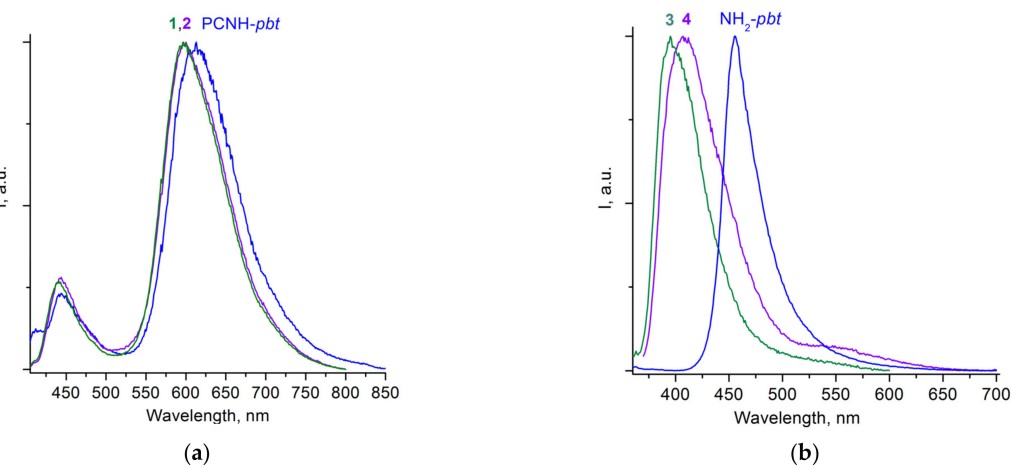

**Figure 3.** Normalized emission spectra of solid samples PCNH-*pbt*, **1** and **2** (**a**) and NH$_2$-*pbt*, **3** and **4** (**b**) at an excitation wavelength of 350 nm.

**Scheme 2.** Excited state proton transfer (ESIPT) in PCNH-*pbt*.

## 3. Materials and Methods

### 3.1. General Procedures

Starting materials were used as received from suppliers. Ethyl acetate was distilled prior to use. IR-spectra were recorded on a Fourier IR spectrometer FT-801 (Simex, Novosibirsk, Russia) in KBr pellets (Figure S8). Elemental analyses were performed on various MICRO cube instruments (Langenselbold, Germany) for C, H, N, and S elements. The content of S was determined with greater uncertainty than for the other elements due to the data processing characteristics of the instrument. The $^1$H NMR spectra were recorded

in $CD_3CN$ using a Bruker DRX-500 spectrometer (Madison, WI, USA) (Figures S9–S11); the solvent peak was used as an internal reference.

Solid emission spectra were recorded at 298 K using an Agilent Cary Eclipse spectrometer (Santa Clara, CA, USA). For the measurements, the samples were placed between two nonfluorescent quartz plates. The spectra of $10^{-5}$ M solutions in 1 cm quartz cuvettes (in the transmittance mode) and of $10^{-3}$ M solutions in 1 mm quartz cuvettes (in the reflection mode) were obtained with the same spectrometer.

### 3.2. X-ray Data

Single-crystal X-ray diffraction (XRD) data for the compounds (Table 1) were collected at 150 K with a Bruker D8 Venture diffractometer equipped with a CMOS PHOTON III detector and IμS 3.0 microfocus source (MoK$\alpha$ radiation ($\lambda$ = 0.71073 Å), collimating Montel mirrors). The crystal structures were solved using SHELXT [29] and were refined using SHELXL [30] programs with OLEX2 GUI [31]. Atomic displacement parameters for non-hydrogen atoms were refined anisotropically. Hydrogen atoms were placed geometrically with the exception for those at the amino groups, which were localized from the residual electron density map and refined with the restrain on N–H bond (0.88 Å). The structures of **2–4** were deposited to the Cambridge Crystallographic Data Centre (CCDC) as a supplementary publication No. 2195999-2196001.

**Table 1.** Crystal data and structure refinement for the compounds.

| Identification Code | 2 | 3 | 4 |
|---|---|---|---|
| Empirical formula | $C_{60}H_{46}Br_2N_4O_4P_2S_2Zn$ | $C_{13}H_{10}Cl_2N_2SZn$ | $C_{13}H_{10}Br_2N_2SZn$ |
| Formula weight | 1238.26 | 362.56 | 451.48 |
| Space group | $P–1$ | $P–1$ | $C2/c$ |
| a/Å | 11.6873(8) | 7.4267(4) | 22.6928(14) |
| b/Å | 16.2088(11) | 8.8927(4) | 11.1268(7) |
| c/Å | 16.5818(9) | 10.7954(6) | 13.1721(7) |
| $\alpha$/° | 71.095(2) | 73.608(2) | 90 |
| $\beta$/° | 69.613(2) | 82.785(2) | 121.735(2) |
| $\gamma$/° | 73.595(2) | 87.943(2) | 90 |
| Volume/Å$^3$ | 2734.2(3) | 678.57(6) | 2828.7(3) |
| Z | 2 | 2 | 8 |
| $\rho_{calc}$ g/cm$^3$ | 1.504 | 1.774 | 2.120 |
| μ/mm$^{-1}$ | 2.097 | 2.341 | 7.519 |
| F(000) | 1256.0 | 364.0 | 1744.0 |
| 2Θ range for data collection/° | 3.312–54.254 | 4.774–61.22 | 4.22–61.034 |
| Index ranges | $-14 \leq h \leq 14, -20 \leq k \leq 20, -21 \leq l \leq 20$ | $-10 \leq h \leq 10, -12 \leq k \leq 12, -15 \leq l \leq 15$ | $-32 \leq h \leq 32, -15 \leq k \leq 15, -18 \leq l \leq 18$ |
| Reflections collected | 33348 | 8165 | 35468 |
| Independent reflections | 11968 [$R_{int}$ = 0.0440, $R_{sigma}$ = 0.0542] | 4129 [$R_{int}$ = 0.0414, $R_{sigma}$ = 0.0589] | 4317 [$R_{int}$ = 0.0459, $R_{sigma}$ = 0.0267] |
| Data/restraints/parameters | 11968/180/828 | 4129/2/178 | 4317/2/178 |
| Goodness-of-fit on F$^2$ | 1.073 | 1.060 | 1.049 |
| Final R indexes [I >= 2σ (I)] | $R_1$ = 0.0511, $wR_2$ = 0.1063 | $R_1$ = 0.0306, $wR_2$ = 0.0675 | $R_1$ = 0.0234, $wR_2$ = 0.0475 |
| Final R indexes [all data] | $R_1$ = 0.0693, $wR_2$ = 0.1127 | $R_1$ = 0.0398, $wR_2$ = 0.0739 | $R_1$ = 0.0310, $wR_2$ = 0.0506 |
| Largest diff. peak/hole/e Å$^{-3}$ | 0.90/−0.57 | 0.40/−0.43 | 0.57/−0.47 |

Powder diffraction patterns (Figures S1–S4, S12 and S13) were recorded using a Stoe STADI MP powder X-ray diffractometer (Darmstadt, Germany) (CuK$\alpha$1 radiation, $\lambda$= 1.54060 Å, generator settings 40 kV/40 mA, curved germanium (111) monochromator, and DECTRIS MYTHEN 1K linear detector (Baden-Taettwil, Switzerland)) in transmission mode (focusing on the detector).

### 3.3. Syntheses
3.3.1. Synthesis of PCNH-*pbt*

The compound was synthesized as reported recently [22]. Under an argon atmosphere, 2-(benzo[d]thiazol-2-yl)aniline (4 mmol) and freshly distilled furfural (4.4 mmol) were dissolved in THF (5 mL). The reaction mixture was stirred at room temperature for 20 min,

followed by the addition of diphenylphosphine (4 mmol, 700 μL). After stirring overnight, all volatiles were removed, and the solid residue was dissolved in 40 mL acetonitrile. Then 1 mL of $H_2O_2$ (30%, 8 mmol) was added, and the reaction mixture was further stirred for 5 days to give a precipitate. The latter was filtered off, washed with acetonitrile, and dried in vacuo. The yield was 82%.

### 3.3.2. Synthesis of [Zn(L)$_2$Hal$_2$] (Hal = Cl, **1**; Hal = Br, **2**)

To a solid PCNH-*pbt* (0.30 mmol) and anhydrous ZnHal$_2$ (0.33 mmol), ethyl acetate (20 mL) was added. The mixture was stirred at an ambient temperature for 4 days. A white precipitate of **1** or **2** was separated by filtration, washed with ethyl acetate, and dried in a vacuum. The yield was 0.034 g (20%) for **1** and 0.073 g (40%) for **2**. The use of a stoichiometric ratio of the reagents (1:2) resulted in a mixture PCNH-*pbt* and the title products with a total mass of the solids of 0.082 g for Hal = Cl (yield of **1** in the mixture is 35%) and 0.092 g for Hal = Br (yield of **2** in the mixture is 26%).

Calc. for **1**, $C_{60}H_{46}Cl_2N_4O_4P_2S_2Zn$ (1149.41): C 62.7, H 4.0, N 4.9, S 5.6. Found C 62.4, H 4.1, N 4.8, S 5.4.

Calc. for **2**, $C_{60}H_{46}Br_2N_4O_4P_2S_2Zn$ (1238.3): C 58.2, H 3.7, N 4.5, S 5.2. Found C 58.1, H 3.8, N 4.5, S 4.8.

IR for **1** (cm$^{-1}$): 563 (s), 595 (m), 695 (s), 736 (s), 819 (w), 886 (w), 929 (m), 962 (m), 1012 (m), 1096 (m), 1121 (m), 1148 (s), 1211 (s), 1284 (w), 1325 (w), 1385 (w), 1438 (s), 1459 (w), 1495 (s), 1524 (s), 1587 (s), 1900 (w), 2921 (w), 3061 (w), 3254 (w), 3452 (w).

IR for **2** (cm$^{-1}$): 563 (s), 596 (s), 694 (s), 735 (s), 819 (s), 886 (w), 928 (s), 962 (s), 1013 (s), 1096 (s), 1121 (s), 1147 (s), 1211 (s), 1260 (w), 1284 (s), 1312 (s), 1325 (m), 1385 (w), 1438 (s), 1459 (s), 1495 (s), 1524 (s), 1587 (s), 1607 (m), 2850 (w), 2919 (w), 3061 (m), 3256 (w), 3452 (w).

### 3.3.3. Synthesis of [Zn(L')Hal$_2$] (Hal = Cl, **3**; Hal = Br, **4**)

To a solid NH$_2$-*pbt* (0.50 mmol) and anhydrous ZnHal$_2$ (0.50 mmol), ethyl acetate (10 mL) was added. The mixture was stirred at an ambient temperature for a week. A pale yellow precipitate of **3** or **4** was separated by filtration, washed with ethyl acetate, and dried in a vacuum. The yield was 0.182 g (90%) for **3** and 0.191 g (85%) for **4**.

Calc. for **3**, $C_{13}H_{10}Cl_2N_2SZn$ (362.59): C 43.1, H 2.8, N 7.7, S 8.8. Found C 42.9, H 2.9, N 7.7, S 8.5.

Calc. for **4**, $C_{13}H_{10}Br_2N_2SZn$ (451.50): C 34.6, H 2.2, N 6.2, S 7.1. Found C 34.6, H 2.3, N 6.2, S 6.7.

IR for **3** (cm$^{-1}$): 577 (w), 630 (m), 659 (w), 725 (s), 755 (s), 805 (s), 877 (w), 951 (s), 987 (s), 1021 (w), 1061 (w), 1078 (s), 1122 (s), 1165 (m), 1179 (s), 1203 (w), 1231 (s), 1255 (w), 1291 (w), 1305 (m), 1324 (s), 1434 (s), 1450 (s), 1486 (s), 1553 (s), 1568 (s), 1605 (s), 1813 (w), 1937 (w), 1979 (w), 3094 (m), 3122 (s), 3161 (s), 3243 (w), 3456 (w), 3754 (w), 3842 (w).

IR for **4** (cm$^{-1}$): 599 (s), 634 (m), 664 (w), 711 (m), 730 (m), 764 (s), 805 (m), 853 (w), 875 (w), 949 (w), 989 (s), 1072 (s), 1114 (w), 1163 (m), 1242 (s), 1294 (m), 1326 (s), 1429 (s), 1462 (s), 1480 (s), 1499 (m), 1566 (s), 1608 (m), 1712 (w), 1808 (w), 1841 (w), 1894 (w), 1932 (w), 1974 (w), 3037 (w), 3067 (w), 3130 (w), 3208 (m), 3272 (m), 3548 (w).

$^1$H NMR for **3** (400 MHz, CD$_3$CN): 8.24–8.20 (m, H), 8.06 (ddd, J = 8, 1.3, 0.6 Hz, H), 7.79 (m, H), 7.55 (dddd, J = 9.2, 8.4, 7.3, 1.2 Hz, 2H), 7.38 (ddd, J = 8.2, 7.2, 1.5 Hz, H), 7.06–6.96 (m, 2H), 6.29 (brs, NH$_2$).

$^1$H NMR for **4** (400 MHz, CD$_3$CN): 8.35 (d, J = 6.4 Hz, H), 8.09 (d, J = 6.4 Hz, H), 7.82 (d, J = 6 Hz, H), 7.62 (t, J = 12.4, 6 Hz, H), 7.53 (t, J = 12.4, 6.4 Hz, H), 7.43 (t, J = 14.8, 6 Hz, H), 7.10 (m, 2H), 6.18 (s, NH$_2$).

$^1$H NMR for NH$_2$-*pbt* (400 MHz, CD$_3$CN): 7.99–7.95 (m, 2H), 7.70 (dd, J = 8, 1.4 Hz, H), 7.50 (ddd, J = 15.6, 8, 0.8 Hz, H), 7.39 (ddd, J = 8.4, 7.1, 1.5 Hz, H), 7.23 (ddd, J = 8.4, 7.1, 1.5 Hz, H), 6.85 (dd, J = 8, 1.2 Hz, H), 6.71 (brs, NH$_2$), 6.70 (dd, J = 8.4, 7.1, 1.5 Hz, H).

## 4. Conclusions

To summarize, we synthesized and characterized novel coordination compounds $[Zn(L)_2Hal_2]$ (Hal = Cl, **1** and Hal = Br, **2**) with the α-aminophosphine oxide comprising a phenylbenzothiazole (*pbt*) moiety. Pure solids **1** and **2** precipitated from the reaction mixture of PCNH-*pbt* and an excess of zinc halide (molar ratio of 1:1). In addition, using the parent $NH_2$-pbt, we synthesized the corresponding Zn compounds, $[Zn(L')Hal_2]$ (Hal = Cl, **3** and Hal = Br, **4**). A comparison of the coordination abilities of $NH_2$-*pbt* and PCNH-*pbt* revealed that the former tends to bind with a metal in a chelate manner via two N atoms. Despite the presence of a number of donor atoms capable of coordination, PCNH-*pbt* (L) binds to Zn only via the oxygen of the P=O group. A steric hindrance of diphenylphosphine and furanyl groups in the neutral ligand is a possible reason why no chelate coordination is observed in complexes **3** and **4**. We assume that deprotonation of PCNH-*pbt* to form the anion PCN-*pbt*$^-$ will increase the donor ability of the N atoms as well as splitting off the $H^+$ while also providing a space for chelate coordination. Remarkably, L′ reveals different spatial geometry in complexes **3** and **4**, featuring a large variation of twist angle between the heterocyclic and aniline moieties. The data demonstrate relatively high flexibility of L′ towards twist along the C–C bond, which is likely tailored by crystal packing effects.

In contrast to free $NH_2$-*pbt* and its coordination compounds **3** and **4**, PCNH-*pbt*, **1** and **2** exhibit dual-band photoluminescence in the solid state. This phenomenon can be rationalized by the following: the minor band at 450 nm is ascribed to a radiative transition for the regular amine species, while the major band at 520–550 nm is associated either with the proton-transferred imine species (ESIPT mechanism) or charge transfer state (TICT) with a different geometry.

**Supplementary Materials:** The following supporting information can be downloaded at: https://www.mdpi.com/article/10.3390/inorganics10090138/s1; Figures S1–S4, S12 and S13, Powder XRD patterns; Figure S5, Hydrogen bond network in **3** and **4**; Figure S6, Excitation spectra of solids PCNH-*pbt*, 1 and 2 on the example of the latter; excitation and emission spectra of PCNH-*pbt*, in tetrahydrofuran and toluene solutions; Figure S7, Excitation and emission spectra of different solid samples of **4**; Figure S8, IR spectra; Figures S9–S11, $^1$H NMR spectra; and Table S1, Torsion angle and hydrogen bond characteristics for compounds comprising NH-*pbt* moiety. Reference [32] is cited in the Supplementary Materials.

**Author Contributions:** Conceptualization, T.S.S.; funding acquisition, T.S.S.; investigation, D.S.K., E.K.P., and T.S.S.; supervision, S.N.K.; visualization, T.S.S.; writing—original draft, T.S.S.; writing—review and editing, S.N.K. All authors have read and agreed to the published version of the manuscript.

**Funding:** This work is supported by the Russian Science Foundation (project no. 21-73-10096). The powder XRD analysis was carried out with the support of the Ministry of Science and Higher Education of the Russian Federation (No. 121031700313-8 and No. 121031700321-3).

**Institutional Review Board Statement:** Not applicable.

**Informed Consent Statement:** Not applicable.

**Data Availability Statement:** CCDC 2195999-2196001 contains the supplementary crystallographic data for this paper. These data can be obtained free of charge via https://www.ccdc.cam.ac.uk/structures (or from the CCDC, 12 Union Road, Cambridge CB2 1EZ, UK; Fax: +44 1223 336033; E-mail: deposit@ccdc.cam.ac.uk).

**Acknowledgments:** Single-crystal X-ray diffraction data were collected with the equipment of the XRD Facility of NIIC SB RAS. Powder X-ray diffraction data were collected with the equipment of the Laboratory of Molecular Design and Ecologically Safe Technologies (MDEST) of the Scientific Educational Center "Institute of Chemical Technology", Novosibirsk State University.

**Conflicts of Interest:** The authors declare no conflict of interest. The funders had no role in the design of the study; in the collection, analyses, or interpretation of data; in the writing of the manuscript; or in the decision to publish the results.

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
