# Peer review of "Luminescent Zn Halide Complexes with 2-(2-Aminophenyl)benzothiazole Derivatives"

_inorganics, doi:10.3390/inorganics10090138_

Round 1

Reviewer 1 Report

This manuscript reports four crystalline compounds based on 2-(2-aminophenyl)benzothi-11 azole (NH2-pbt) and its phosphorus-containing derivative and their photoluminescence properties have been investigated. I would like to see it publish in this journal, and before acceptance some minor revisions are still needed.

1. Remove “novel” from title.

2. Double check the wavelength for CuK-alpha or CuK-alpha1.

3. A space is needed in cases like “150K”. Please carefully check the whole manuscript.

4. The excitation spectra for the resultant compounds should be added.

5. The authors state that the shoulder emission is associated with ESIPT mechanism. Could the authors give some experimental evidence; for instance, please check whether the compounds could still exhibit dual emission under basic or acidic conditions by breaking the ESIPT process.

6. The color of balls corresponding to each element should be presented in Figures 1 and 2 for better reading.

7. I suggest the description of crystal structures in the manuscript following the sequence of compounds 1, 2 and then compounds 3 and 4.

8. In caption of Figure 2, an “and” should be added to “disorder is omitted, N–H···N hydrogen bond is marked 176 by dashed red line.” after the comma. Same for Figure 2b. Please double check the problem throughout the whole manuscript.

9. It is better to mark the graph sets in Figures.

10. In Figure 3, the compound numbers should be bolded.

11. Since the major content of this paper is about crystal structure, I would suggest that the crystallographic data be presented in main text. Selected bond lengths and angles could also be added to the structural figures.

Author Response

We are grateful to the Reviewer for the helpful comments and for giving us the opportunity to improve the manuscript. We considered all the suggestions and revised the text accordingly (the changes are marked yellow). The point-by-point responses are given below.

  1. Remove “novel” from title.

We have revised the title accordingly

  1. Double check the wavelength for CuK-alpha or CuK-alpha1.

We have revised the wavelength in the main text and in the supporting information (it is CuK-alpha1).

  1. A space is needed in cases like “150K”. Please carefully check the whole manuscript.

We have revised the text accordingly.

  1. The excitation spectra for the resultant compounds should be added.

We have added the corresponding spectra in the supporting information (Figs. S12-13)

  1. The authors state that the shoulder emission is associated with ESIPT mechanism. Could the authors give some experimental evidence; for instance, please check whether the compounds could still exhibit dual emission under basic or acidic conditions by breaking the ESIPT process.

Unfortunately, compounds 1 and 2 are poorly soluble in organic solvents, and thus we cannot study their properties in the solution. However, we tested the emission of PCNH-pbt that shows similar dual band behaviour. Interestingly, solutions of PCNH-pbt in tetrahydrofuran (2·10–5 M and 10–3 M) and toluene (2·10–5 M) reveal only the short wavelength band (Fig. S12b). Addition of an excess of triethylamine does not change the emission. Thus, the long wavelength is attributed to some solid-state effects. Since no strong intermolecular interactions of the same nature are observed in crystals of PCNH-pbt, 1, and 2, we can assume that the long wavelength band appears in the solid PCNH-pbt owing to geometry differences of the molecule in the solid state and solution. The restriction of the intramolecular motion (RIM) in the solid could also be responsible for the band appearance. We have included this discussion in the main text.

  1. The colour of balls corresponding to each element should be presented in Figures 1 and 2 for better reading.

We have included atomic colour scheme in the figure captions.

  1. I suggest the description of crystal structures in the manuscript following the sequence of compounds 12and then compounds 3and 4.

We believe that the current sequence is more appropriate, since L’ is parent to L. Thus, we would like to keep the current version of the text.

  1. In caption of Figure 2, an “and” should be added to “disorder is omitted, N–H···N hydrogen bond is marked 176 by dashed red line.” after the comma. Same for Figure 2b. Please double check the problem throughout the whole manuscript.

We have revised the text accordingly

  1. It is better to mark the graph sets in Figures.

We have marked the graph sets in Figure S11.

  1. In Figure 3, the compound numbers should be bolded.

We have revised the figure accordingly

  1. Since the major content of this paper is about crystal structure, I would suggest that the crystallographic data be presented in main text. Selected bond lengths and angles could also be added to the structural figures.

We have included Table S1 as Table 1 in the main text and included selected bond lengths in the caption to the figures.

Reviewer 2 Report

The research work of Taisiya Sukhikh and co-authors focuses on the synthesis and comparison of coordination behavior of four Zn-halide complexes bearing either 2-(2-aminophenyl)benzothiazole or its phosphine oxide analog. Moreover, the report is accompanied by photophysical studies recorded for solid samples. No doubt, photoluminescent Zn complexes have been highly attractive in the last decades due to their potential application in security tracker devices, sensors, and emitting devices. However, publication in the current view cannot be suggested for publishing without further revision. Whereas the crystallographic part sounds well, the authors weren’t able to provide a clear idea of the purity of the sample. Thus, in the next round, I will be happy to see NMR data for all complexes attached to the ESI, including non-characterized by this method complexes 1 and 2. Considering the presence of the phosphorus nuclei, 31P spectra also should be provided. Moreover, as elemental analysis of none of the complexes is given, it is strongly recommended to fulfill the existing gap.The authors described the origin of the dual emission (complexes 3,4) as a result of ESIPT or CT, however, the lack of photophysical characterization keeps discussion only as a shallow assumption. Provided steady-state spectra are not representative and cannot be treated as accurate. As a hint, the authors could record and analyze excitation spectra together with time-resolved data of solids and solutions. A significant deviation of the emission intensities upon different photoexcitation of complexes 1,2 also remains the purity question open (line 190).

Further minor comments for consideration:

1.     Yields, conditions, and solvents on Scheme 1 are missing.

2.     IR description is expected to be modified prior to further submission (band (intensity), i.e. 1923 cm-1 (w)). 

3.     Some important bonding or angle information is missing, such as Zn – halide, Zn – O-P, etc. 

4.     Line 90 – Hal = br – Br.

5.     Line 162  "Two ligands.." - Both ligands

6.     Line 192 –  “The emission spectra exhibit a single band at 395 and 407 nm for 3 and 4, respectively,…” The emission spectra consist of two distinct maxima, but not one. Excitation spectra on both maxima are also requested from the authors. 

7.     Moreover, I strongly suggest removing reference 25 from the reference list as it does not contain any reliable information, or to the date, it is missing. Instead, the number of the structure, if applicable, (CCDC database) of the free ligand can be cited.

Author Response

We are grateful to the Reviewer for the helpful comments and for giving us the opportunity to improve the manuscript. We considered all the suggestions and revised the text accordingly (the changes are marked yellow). The point-by-point responses are given below.

I will be happy to see NMR data for all complexes attached to the ESI, including non-characterized by this method complexes 1 and 2. Considering the presence of the phosphorus nuclei, 31P spectra also should be provided. Moreover, as elemental analysis of none of the complexes is given, it is strongly recommended to fulfill the existing gap. The authors described the origin of the dual emission (complexes 3,4) as a result of ESIPT or CT, however, the lack of photophysical characterization keeps discussion only as a shallow assumption. Provided steady-state spectra are not representative and cannot be treated as accurate. As a hint, the authors could record and analyze excitation spectra together with time-resolved data of solids and solutions. A significant deviation of the emission intensities upon different photoexcitation of complexes 1,2 also remains the purity question open (line 190).

We have included figures of the NMR spectra of NH2-pbt, 3 and 4 in the ESI. Unfortunately, once solids 1 and 2 precipitated from the mother liquor, they become poorly soluble in organic solvents, which does not allow us to study them by means of NMR spectroscopy. This information has been included in the text.

Results of the elemental analysis have been included in the experimental section. The data agree well with the calculated content.

The study of emission mechanisms and purity problems are indeed complicated questions. The data of PXRD and elemental analysis, as well as NMR spectroscopy reveal no impurities within the sensitivity of the methods. Emission spectroscopy is much more sensitive method, so we cannot exclude the probability of detection of very minor impurities.

We measured the excitation spectra (Figs. S12 and 13); unfortunately, we cannot carry out time-resolved experiments in the nanosecond range at this moment. Preliminary study after rime delay of 100 μs revealed that compounds 3 and 4 feature relatively short emission lifetimes, while the minor band of compounds 1 and 2 exhibits a phosphorescent nature. In addition, we have studied the emission of PCNH-pbt in the solution, which revealed only the short wavelength band. Thus, the long wavelength is attributed to some solid-state effects. Since no strong intermolecular interactions of the same nature are observed in crystals of PCNH-pbt, 1, and 2, we can assume that the long wavelength band appears in the solid PCNH-pbt owing to geometry differences of the molecule in the solid state and solution. The restriction of the intramolecular motion (RIM) in the solid could also be responsible for the band appearance. Definitely, the results do not directly indicate the presence ESIPT or TICT, but they do not contradict with each other.

Further minor comments for consideration:

  1. Yields, conditions, and solvents on Scheme 1 are missing.

We would like to keep the current version in order not to overcomplicate the scheme. The corresponding data are indicated in the experimental section.

  1. IR description is expected to be modified prior to further submission (band (intensity), i.e. 1923 cm-1 (w)).

We have modified the data accordingly

  1. Some important bonding or angle information is missing, such as Zn – halide, Zn – O-P, etc. 

We have included the corresponding values in the caption to Figs. 1 and 2.

  1. Line 90 – Hal = br – Br.

We have revised the text accordingly

  1. Line 162 – "Two ligands.." - Both ligands

We have revised the text accordingly

  1. Line 192 –  “The emission spectra exhibit a single band at 395 and 407 nm for 3 and 4, respectively,…” The emission spectra consist of two distinct maxima, but not one. Excitation spectra on both maxima are also requested from the authors. 

We have revised the text and have included the excitation spectra as Fig. S13b.

  1. Moreover, I strongly suggest removing reference 25 from the reference list as it does not contain any reliable information, or to the date, it is missing. Instead, the number of the structure, if applicable, (CCDC database) of the free ligand can be cited.

Reference 25 corresponds to the accepted article, which is going to be published in the next issue of the Journal of Structural Chemistry. The Journal is translated in English and is published by Springer (https://www.springer.com/journal/10947), so a reader can soon access to the article. The CCDC refcode is clarified in the footnote to Table S1, which we refer to when discussing the structure of PCNH-pbt.

Round 2

Reviewer 2 Report

After reading the revisited manuscript, I recommend the paper be published in Inorganics without further correction.